# Unsupervised Federated Learning for Face Recognition in Decentralized Environments

## Abstract

Recent advancements in face recognition involve training on a single computer, often containing sensitive personal information, raising privacy concerns. To address this, attention turns to federated learning for unsupervised face recognition, leveraging decentralized edge devices. Each device independently undergoes model training, transmitting results to a secure aggregator. We utilize GANs to diversify data without the need for transmission, thereby preserving privacy throughout the entire process. The aggregator integrates these diverse models into a single global model, which is then transmitted back to the edge devices for continued improvement. Experiments on CelebA datasets demonstrate that federated learning not only preserves privacy but also maintains high levels of performance.

## 1 Introduction

Facial recognition, an automated procedure that recognizes facial features to recognize or authenticate individuals, has become indispensable in numerous security and authentication systems Zhao et al. (2003).

The amalgamation of ML advancements with the accessibility of facial data has notably boosted the accuracy and efficacy of face recognition systems. Mostly, these systems leverage ML methodologies to train deep neural networks using facial dataset primarily gathered from end-devices such as smartphones, with centralized model training on servers Chen & Ran (2019). Nonetheless, this configuration gives rise to concerns regarding privacy and strains on communication infrastructure.

Privacy apprehensions represent a significant challenge in face and speaker recognition systems Woubie & Bäckström (2021); Solomon et al. (2023a); Woubie et al. (2023); Solomon et al. (2023c;b); Woubie et al. (2024), given the customary complete sharing of facial data, which can pose serious threats to privacy.

Federated learning(FL) offers a solution by advocating for a distributed training approach. Rather than dispatching raw sensitive data to a central server for training, each device retains its own model and trains it using local data McMahan et al. (2017). Only model updates are then sent to the central server, which aggregates and integrates them into the global model.

As mobile devices generate vast amounts of data daily, FL emerged as a solution to maintain data on the device, shifting the network's focus to the edge. Numerous companies have embraced FL, highlighting its significance in privacy-sensitive applications, particularly when training data is distributed across devices Li et al. (2020). This demand surge has prompted the development of various tools, signaling increasing interest in privacy-preserving techniques Beutel et al. (2020); Yang (2021).

Privacy concerns loom large over face and speaker recognition systems, as they often entail the sharing of sensitive facial data, posing a threat to privacy. However, FL emerges as a beacon of hope, offering a solution by facilitating model training directly on user devices. This approach ensures that sensitive data stays local, bolstering privacy and minimizing the necessity for data transmission.

This work explores incorporating FL techniques into the training of deep neural network-based face recognition classifiers to protect user privacy. The proposed system allows each device to independently train its model and send it to a secure aggregator or central server, ensuring privacy

and efficient model training. Moreover, using GANs on edge devices eliminates the need to transmit synthetic data, further boosting privacy and efficiency.

The different applications of FL are many, spanning smartphone-based learning and collaborative learning across organizations. These systems offer confidentiality while delivering promising results in facial recognition tasks, providing a quantitative understanding of the privacy-accuracy trade-off.

## 2 Related Work

### 2.1 Preserving Privacy

A variety of methods have been developed to enhance the privacy and security of facial data. One innovative technique, described in Bellovin et al. (2019), involves generating synthetic images instead of using real facial images. This is accomplished by training a class-conditional GAN, where the generator learns from the original dataset and uses identity labels as class markers. These synthetic images are then used to train the face recognition system. Another approach in Arman et al. (2024), enhances privacy by applying locality-sensitive hashing, which adds randomness to facial data to prevent unauthorized use or reconstruction. Additionally, Boulemtafes et al. (2020) presents a method using the Householder matrix to protect both models and facial data, combining additive and multiplicative perturbations to streamline user-side processing.

The authors in Gupta et al. (2024) proposes encrypting facial images with affine transformations, including permutation, diffusion, and shift transformations, to preserve privacy. Another approach, detailed in Wang et al. (2022), focuses on frequency domain privacy-preserving face recognition. This technique collects same-frequency components from different blocks using an analysis network and applies a fast masking technique to secure the remaining frequency components. Additionally, Seid (2024) describes the encryption of normalized face feature vectors using the CKKS algorithm from the SEAL library for enhanced security.

Furthermore, several studies employ local differential privacy methods to prevent the reverse-engineering or identification of specific data points. As an example, Yao et al. (2024) offers a robust privacy protection framework for face recognition systems operating at the edge. This framework uses a local differential privacy algorithm based on feature information proportion differences. Moreover, it incorporates identity authentication and hash techniques to validate terminal devices and ensure the integrity of face images during data capture.

### 2.2 Face Recognition via Federated Learning

Several techniques employ FL to enhance privacy in face recognition applications. One such method highlighted in Xu et al. (2023), which uses privacy-agnostic clusters during the training phase. These clusters are structured to protect sensitive personal information. PrivacyFace consists of two primary elements: the Differently Private Local Clustering (DPLC) algorithm, which identifies group features that are independent of privacy, and a consensus-aware loss function for face recognition that improves the distribution of the global feature space by utilizing these non-sensitive group features.

As outlined in Li et al. (2021), it introduces a FL framework specifically designed to learn from face images across multiple clients while preserving data privacy. In this method, each client, typically a mobile device, exclusively holds face images belonging to its respective owner. By ensuring that the images are not shared with other clients or a central host, FedFace maintains the privacy of individual user data while enabling collaborative learning for face recognition tasks.

The authors in Shao et al. (2022), strives to create generalized fPAD (Face Presentation Attack Detection) models while maintaining data privacy. This approach involves each data owner training a local model, with a server subsequently aggregating these models without accessing individual private data. Following refinement of the global model, it is then deployed for fPAD inference, ensuring both effectiveness and privacy in face presentation attack detection.

FedFR introduces a FL framework dedicated to privacy-aware generic face representation. This innovative framework prioritizes the optimization of personalized models for clients through the utilization of the Decoupled Feature Customization module. Through this approach, FedFR not

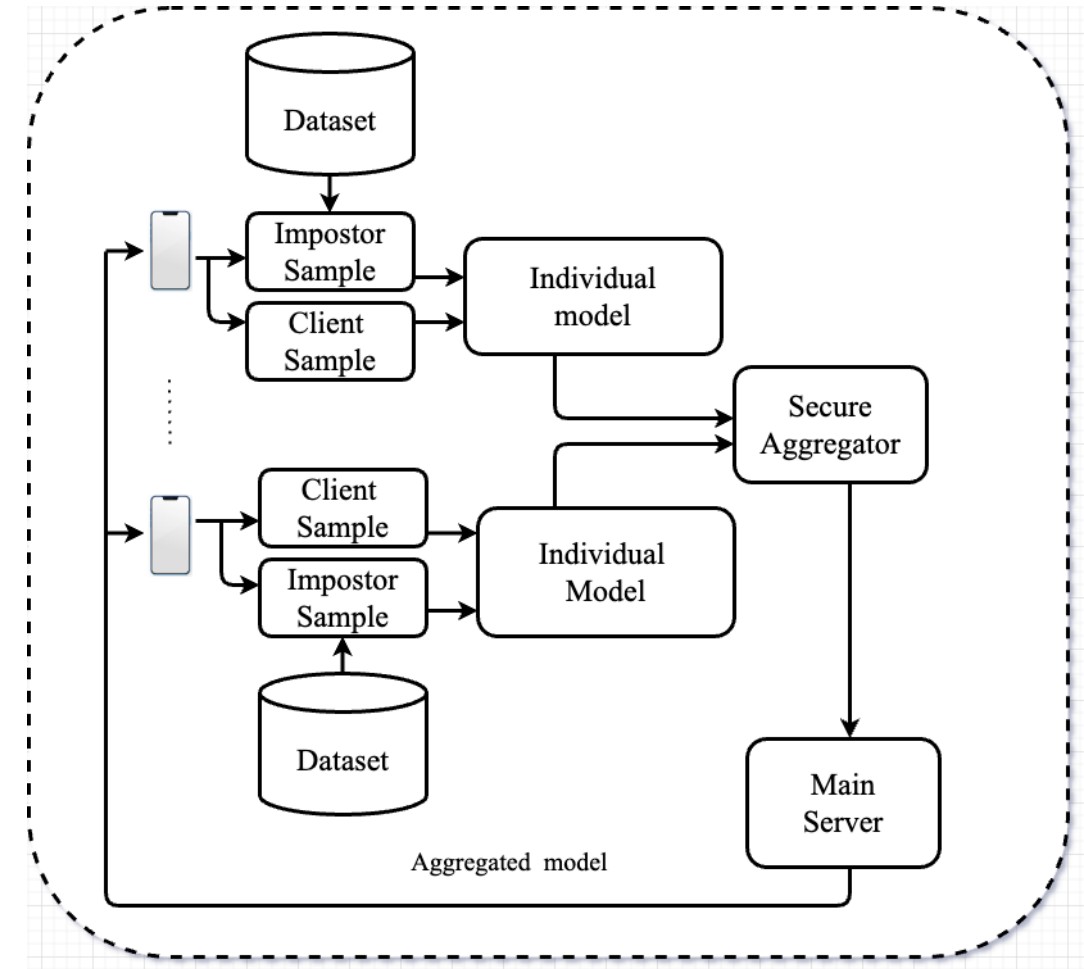

Figure 1: The proposed face recognition system integrates federated learning. By employing a secure aggregator, we enable a network of inherently untrusting devices to collaborate and compute an aggregated value without revealing their individual private data.

only improves the global model for face representation but also tailors the personalized user model to enhance overall performance and privacy preservation.

## 3 PROPOSED SYSTEM

Creating a cohesive global statistical model from data dispersed across a restricted or extensive array of remote devices presents a daunting hurdle in FL. Commonly, the central aim in FL is to minimize the subsequent objective function.

As depicted in Fig. 1, the FL system operates across three key locations: edge devices, a secure aggregator, and a central main server.

In the realm of FL, the challenge lies in training a centralized model using a distributed dataset, where numerous nodes, akin to user devices, possess subsets of data with varying sizes. These nodes compute individual model updates at the device level, subsequently transmitting them to a central server. In each training iteration, a significant number of these updates or gradients are gathered by the central server, which then calculates a global update by averaging the individual local updates.

While training individual face recognition models in an unsupervised system using only client images from a specific device is possible, our goal is to enhance model robustness and improve the differentiation of impostor images. To achieve this, as shown in Figure 1, two distinct methods are employed to generate impostor image data for each individual on the edge device. In the first

method, impostor image data for a given person is randomly selected from images of other individuals in the CelebA dataset. In the second method, a GAN model is trained to generate impostor image data, as it's not always viable to acquire image data of different individuals on edge devices. We trained a GAN model using CelebA. After generating the impostor images with the trained GAN, they are combined with client image data to train an the face recognition model on the edge device.

In our proposed system, the distributed gradient descent used as as the mechanism to training a deep neural network across training data stored on user-held devices, thereby facilitating an evaluation of the effects of a secure aggregator. In the system incorporating a secure aggregator, the process unfolds with distinct steps, including Local Training, Model Transmission to Secure Aggregator, Global Model Creation, Aggregated Model Transmission, and Distribution to Devices. Conversely, in the absence of a secure aggregator, the workflow proceeds similarly, albeit without involving the aggregator.

Privacy considerations stand at the forefront of driving the adoption of FL applications. These systems prioritize the exchange of model updates, like gradient information, over raw and potentially sensitive data, thus bolstering data privacy while enabling training of a robust models without leaking information to undue risks or breaches. Although FL reduces some privacy risks by abstaining from direct sharing of raw data, it's crucial to recognize that transmitting model updates during the training process can still present potential privacy challenges He et al. (2017). Recent advancements in FL, such as secure multiparty computation (SMC) or differential privacy (DP), have made strides in enhancing privacy. However, these approaches entail trade-offs between privacy and model performance. The secure aggregator, a component of secure multi-party computation algorithms, ensures the privacy and confidentiality of individual model updates while facilitating collaborative computation.

To protect the privacy of federated learning, our proposed system integrates secure multiparty computation (SMC) techniques, as outlined in Choi & Butler (2019). Secure multiparty computation ensures the privacy and confidentiality of individual model updates. This entails local optimization carried out by participating clients, followed by a server step for updating the global model. To tackle the challenge of transferring substantial volumes of updated model parameters from users to a server—often restricted by throughput—strategies like minimizing the number of active users can be utilized. This can be achieved through the implementation of effective scheduling policies.

## 4 EXPERIMENTS

### 4.1 DATASET

Within the fields of computer vision and machine learning, CelebA (Celebrities Attributes Dataset) Liu et al. (2015) emerges as a significant resource. CelebA serves as a cornerstone for numerous endeavors in facial recognition and image analysis. It features an extensive collection of over 200,000 images of celebrities from diverse backgrounds and professions. Each image in the CelebA dataset is carefully annotated with a comprehensive set of 40 binary attributes. These annotations are essential for tasks such as facial attribute prediction and manipulation. Additionally, CelebA provides identity labels for the depicted celebrities, enhancing its utility for face recognition tasks. Renowned for its extensive coverage of poses, expressions, lighting conditions, and backgrounds, CelebA is exceptionally well-suited for a wide spectrum of computer vision applications. Typically partitioned into training, validation, and test sets, the dataset facilitates seamless model training and evaluation processes.

### 4.2 EXPERIMENTAL SETUP

In our approach, we use a convolutional neural network (CNN) architecture that closely resembles VGG-M Chatfield et al. (2014), which is renowned for its effectiveness in tasks ranging from image classification to speech technology. We also integrate a max-pooling layer of of 2 by 2. We also applied batch normalization and dropout layers.

For model training, we rely on the Keras deep learning library Chollet (2021). The training process unfolds on Titan X GPUs, spanning for 100 epochs with a batch size of 64. Our training methodology involves stochastic gradient descent (SGD) with momentum (0.9), incorporating weight decay

Table 1: Equal Error Rate (EER) of individual and federated methods, with and without the use of a Secure Aggregator.

| Proposed Method | With Secure Aggregator | Without Secure Aggregator |
|---|---|---|
| Individual federated Model | | 2.66 |
| Federated federated Model | 3.16 | 2.44 |

$(5E - 4)$ and utilizing a logarithmically decaying learning rate. The learning rate is initialized at $10^{-2}$ and diminishes to $10^{-8}$.

In our work, we subjected the face verification system to rigorous evaluation utilizing CelebA Liu et al. (2015), a well-established and widely used database in the field. From this dataset, we randomly selected 1000 persons' face images, partitioning them such that 90% were allocated for training personalized face models, while the 10% were used for evaluation purposes. If there are 100 face images in the development set, 90 images are allocated for training, while the remaining 10 are reserved for evaluation. To ensure a comprehensive assessment, impostor data was introduced into the test set.

Our approach was focused on training individual face models solely with authentic client face data. Nonetheless, due to the restricted number of files per individual, resulting in most individuals having fewer than 100 face images, this method resulted in overfitting. Consequently, we adjusted our strategy to incorporate both authentic and impostor face data during model training.

To generate impostor images on each individual edge device, we adopted two distinct methods. Initially, we selected face images of other individuals from the dataset as impostor face images, with 100 samples chosen for each device. Subsequently, we employed a GAN model to generate impostor data. Similar to the first method, 100 impostor face images were generated for each device.

One significant challenge encountered during the training of the GAN model lies in its time-consuming nature. Training the GAN model for 50 hours on the CelebA dataset with a Quadro P2000 GPU incurs a computational cost equivalent to 3.5 hours. Nevertheless, once trained, the GAN model facilitates rapid extraction of impostor face image samples on edge devices. It's important to note that the GAN model training is a one-time task.

To assess the effectiveness of our system, we employed the Equal Error Rate (EER) metric. The EER represents the point at which the acceptance and rejection error rates are equal, offering valuable insights into the system's overall performance.

### 4.3 RESULTS

As outlined in Section 3, our study analyzes the impact of FL on unsupervised face verification systems, examining scenarios with and without the use of a secure aggregator. Consequently, we present below the experimental findings of the unsupervised system with and without the implementation of the secure aggregator. In Fig.2 (a), approximately 681 devices achieve an EER below 2.34.

This visual representation demonstrates a significant improvement in EER when transitioning from individual models to federated models without a secure aggregator in the unsupervised system. The collaborative approach appears to enhance the performance of the face image models, resulting in lower EER values for a considerable number of devices.

In contrast to the results shown in Fig. 2 (b), this highlights the crucial need to carefully evaluate the impact of a secure aggregator on EER results, indicating a possible trade-off between privacy-enhancing measures and model performance within the unsupervised system.

Table 1 presents the mean Equal Error Rate (EER) for individual models across the 1,000 subjects in the unsupervised face verification system, which is recorded at 2.57. In contrast, the table shows that the average EER for the 1,000 devices in the federated model, within the same system but without a secure aggregator, is 2.36. This reflects an 8.57% relative improvement in EER compared to the baseline unsupervised system. However, it is important to note that incorporating a secure

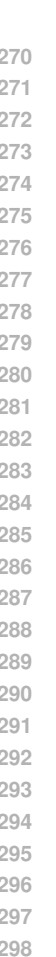

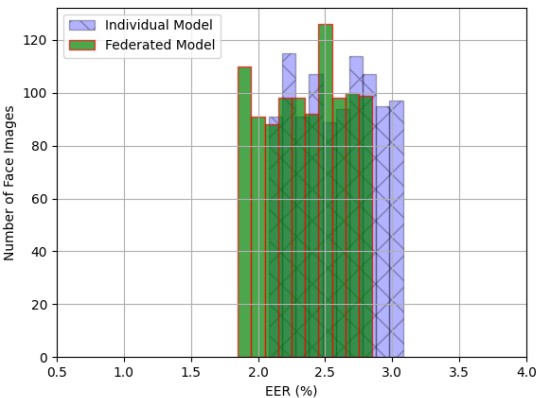

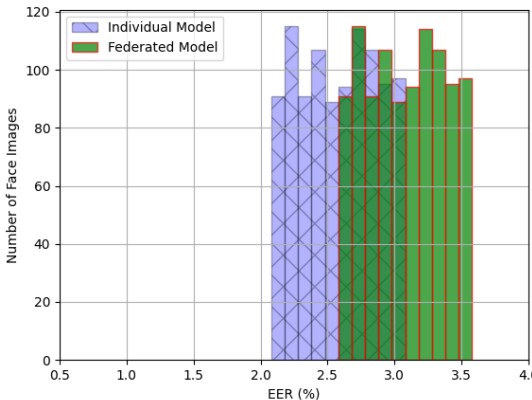

Figure 2: EER of individual and federated models (a) without secure federated aggregator and (b) with secure federated aggregator respectively.

aggregator in the federated model results in a less favorable outcome than the baseline system, as indicated in the table.

## 4.4 DISCUSSIONS

The results presented in Fig. 3 highlight a clear trend: incorporating a secure aggregator often reduces the performance of the federated model. In contrast, when the model functions without a secure aggregator, it demonstrates enhanced EER results compared to individual models. While there is a slight decline in performance associated with the secure aggregator, its role in maintaining data privacy is vital.

Despite this minor decrease in EER with the secure aggregator, the overall performance remains acceptable. This slight compromise is balanced by the privacy advantages that the aggregator provides. The satisfactory EER results, even with its inclusion, emphasize the delicate balance between privacy protection and model efficacy.

Furthermore, we conducted an experiment where all face image samples from 1,000 individuals were combined to train a single generic face recognition model on a standalone computer. The results showed an average EER of 2.8%, which is comparable to that of the federated model (see Fig. 3). This indicates that the federated model can achieve similar EER values while safeguarding the privacy of facial image data, showcasing the benefits of FL in face recognition applications.

The work also compares individual models with federated models across 1,000 devices. Statistical analysis using Student's t-test confirms the significance of the observed differences. The P-values for

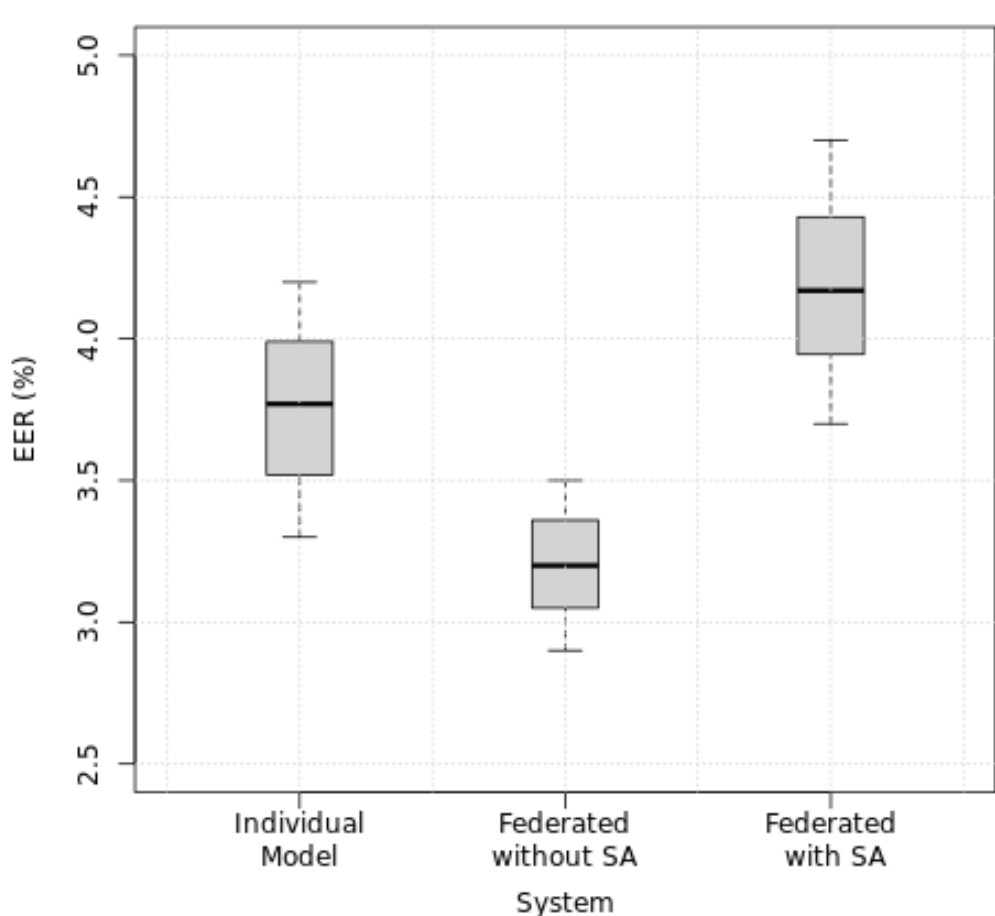

Figure 3: Using 1000 devices the distribution of equal error rate for single and federated models

both comparisons where the federated model uses impostor images from CelebA (federated model 1) and those generated by GAN (federated model 2) are below the typical significance threshold of 0.05. We conclude that the null hypothesis can be rejected. This shows that that the mean EER differences of the individual and federated models are statistically significant.

Lastly, we investigated the effects of updating local models multiple times. The results suggest that frequent updates do not improve EER performance, likely due to the similarity of training data across devices in each training round. Even though increasing the update frequency did not improve performance, this decision was made to prioritize data privacy.

## 5 CONCLUSIONS

This study examines the application of federated learning to bolster the privacy of facial image data on edge devices, specifically in recognition systems. Our approach focuses on decentralized training, thereby avoiding the need to send raw image data to central servers. Instead, each user's data is securely handled on their own edge device. Training occurs locally, with updates from each device contributing to a central model. A secure aggregator then integrates these local models into a federated version, which is sent back to the individual devices via the main server. We also evaluate the impact of the secure aggregator on the face recognition system's performance.

The proposed system presents two main benefits: it effectively safeguards the privacy of facial images by keeping raw data on the user's device. Our experiments show that the federated model, when run without the secure aggregator, achieves a significantly lower average Equal Error Rate (EER) than the individual models. However, including the secure aggregator results in a minor reduction in the aggregated model's EER compared to individual models. These outcomes emphasize the critical balance between maintaining privacy and optimizing performance.

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
