# OpenReview forum: "Unsupervised Federated Learning for Privacy Preserving in Face Recognition System"
_ICLR.cc/2025/Conference — Submitted to ICLR 2025_

### Official Review · Reviewer_t7jk · 2024-10-22

**Soundness:** 1
**Presentation:** 1
**Contribution:** 1
**Rating:** 1
**Confidence:** 4

**Summary:**

This paper introduces an approach to face recognition that leverages federated learning and GANs to address critical privacy concerns. By implementing a decentralized architecture where edge devices independently train models and communicate only processed results to a secure aggregator, the system can ensure sensitive personal data remains protected throughout the entire process.

**Strengths:**

The paper addresses an important topic on face recognition in federated learning.

**Weaknesses:**

1. The contribution in this paper is not clear. The paragraphs in the Introduction section lacks logic flow about the intuition, challenges and contribution of the work.
2. In Section 3, the proposed method is not sufficiently explained. Figure 1 gives the whole framework, but it lacks comparison with prior research and why this framework is better than prior research. It seems it makes no difference from a traditional FL framework.
3. The experiments do not support the effectiveness of the method. Only one dataset and one model is evaluated.
4. Which GAN model is used in the work? It is not clearly mentioned in the paper.
5. It seems the paper is not a finished work. The writing, presentation of the method, and experiments are way below the ICLR standard.

**Questions:**

See weaknesses above.

---

### Official Review · Reviewer_vnQU · 2024-10-23

**Soundness:** 1
**Presentation:** 1
**Contribution:** 1
**Rating:** 3
**Confidence:** 5

**Summary:**

The authors aim to explore the privacy issues in federated learning. The idea is to employ GANs to diversify data locally, eliminating the need for data transmission and thus preserving privacy throughout the process. The aggregator combines these diverse models into a single global model, which is then distributed back to the edge devices for further enhancement.

**Strengths:**

+ This work considers a practical scenario- a face verification system, which is meaningful for exploring privacy issues.

**Weaknesses:**

However, this work has the following major concerns:

**Lack of Novelty and Technical Contribution**: The idea of using GAN to generate impostor images is very straightforward. As explained on Page 4, "In the system incorporating a secure aggregator, the process unfolds with distinct steps, including Local Training, Model Transmission to Secure Aggregator, Global Model Creation, Aggregated Model Transmission, and Distribution to Devices. Conversely, in the absence of a secure aggregator, the workflow proceeds similarly, albeit without involving the aggregator." The core idea is simply using secure aggregation, which is not novel. Recently, a lot of works improve the secure-aggregation-based federated learning for the goal of efficiency or scecurity.  For example,

[Reference 1] FedCSCD-GAN: A secure and collaborative framework for clinical cancer diagnosis via optimized federated learning and GAN.

[Reference 2] Ifl-gan: Improved federated learning generative adversarial network with maximum mean discrepancy model aggregation.

[Reference 3] Scionfl: Efficient and robust secure quantized aggregation

[Reference 4] Cryptography-inspired federated learning for generative adversarial networks and meta-learning.

[Reference 5] Sear: Secure and efficient aggregation for byzantine-robust federated learning

[Reference 6] Efficient Aggregation of Face Embeddings for Decentralized Face Recognition Deployments.

[Reference 7] A comprehensive experimental comparison of the aggregation techniques for face recognition

Moreover, secure aggregation with GAN has been proposed in reference 4, which supports stronger security than this work.

 **Insufficient Experimental Validation**: This work shows experiments conducted on the CelebA dataset. However, it does not provide a sufficient evaluation of other datasets (e.g.,  Wild (LFW) and YouTube Faces (YTF) datasets) or compare their work with other advanced works. The experimental results are very limited. This work lacks sufficient experimental results compared with related works (e.g., reference 3, reference 4, reference 6, and reference 7 as mentioned) that improve secure aggregation or federated GAN.

**Lack of Security Analysis** The security model is not clear and not defined. Accordingly, the formal security analysis also remains lacking. There is no explanation or assumption of the attackers, e.g., the attacker's knowledge, malicious or honest. The explicit privacy guarantee is also not defined. Additionally, secure aggregation is not secure againset some attacks, such as,

[reference 8] Secure Aggregation is Not Private Against Membership Inference Attacks.

**Severe Wring issues.** The introduction does not give a clear summary of motivation and contribution to this work, such as improvement to prior works. For example, the work does not explain why not adopt the better secure aggregation (for example, reference 3) for improved efficiency. The related works of secure aggregation and federated GAN lacks. The explicit motivation to adopt secure aggregation and federated GAN in face recognization remains lacking. Besides, the figure size is weird, e.g, Figure 1 and Figure 3. The writing quality should be hugely improved to meet the paper's qualifications.

**Questions:**

Please refer to weaknesses.

---

### Official Review · Reviewer_nzCT · 2024-11-02

**Soundness:** 2
**Presentation:** 2
**Contribution:** 2
**Rating:** 3
**Confidence:** 5

**Summary:**

The paper employs a Federated Learning (FL) framework by independently training models on edge devices and sending updates to a central server instead of directly transmitting sensitive raw data. Additionally, it uses Generative Adversarial Networks on edge devices to generate diversified data to enhance privacy protection.

**Strengths:**

To improve model robustness, the paper proposes two methods for generating spoofed image data on edge devices: randomly selecting images of other individuals from the CelebA dataset and using GAN models to generate images.

**Weaknesses:**

1- Lack of innovation. The proposed method is a combination of several existing methods, and the innovative aspects are insufficient to support acceptance by the conference.
2- Insufficient experiments. There are too few metrics for measuring facial recognition accuracy, and it is suggested to refer to the literature by Wang et al. (2022). The dataset used is singular, and it is hoped that additional training and testing datasets will be included.

**Questions:**

Please see Weaknesses.

---

### Official Review · Reviewer_TZEx · 2024-11-04

**Soundness:** 1
**Presentation:** 1
**Contribution:** 1
**Rating:** 1
**Confidence:** 5

**Summary:**

The paper proposes a privacy-focused face recognition method using federated learning and GANs, where edge devices train models locally and share only processed data with a secure aggregator.

**Strengths:**

The topic is important.

**Weaknesses:**

The paper is overall poorly presented.
1. The contributions are not clearly stated.
2. The proposed method is just a very simple federated learning process. No innovations at all.
3. Experimental results are very simple and do not provide any insights.

**Questions:**

See the weakness section.

---

### Meta-Review · Area_Chair_YQTF · 2024-12-21

**Metareview:**

Considering the high standards for ICLR publications, this submission falls clearly below the bar for acceptance. All reviewers agree that the paper's quality is poor, with limited technical contributions and insufficient novelty in the proposed method. Additionally, the writing lacks polish. Overall, the submission does not meet the criteria expected at ICLR.

**Additional Comments On Reviewer Discussion:**

The authors did not rebuttal.

---

### Decision · Program_Chairs · 2025-01-22

Reject